# COVID-19 Vaccine Safety Monitoring Studies in Low- and Middle-Income Countries (LMICs)—A Systematic Review of Study Designs and Methods

**DOI:** 10.3390/vaccines11061035

**Published:** 2023-05-29

**Authors:** Malede Mequanent Sisay, Camila Montesinos-Guevara, Alhadi Khogali Osman, Putri Widi Saraswati, Binyam Tilahun, Tadesse Awoke Ayele, Fariba Ahmadizar, Carlos E. Durán, Miriam C. J. M. Sturkenboom, Peter van de Ven, Daniel Weibel

**Affiliations:** 1Department of Data Science and Biostatistics, Julius Center for Health Sciences and Primary Care, University Medical Center Utrecht, 3584 CG Utrecht, The Netherlands; 2Centro de Investigación en Epidemiología Clínica y Salud Pública (CISPEC), Facultad de Ciencias de la Salud Eugenio Espejo, Universidad UTE, Quito 341113, Ecuador; 3Department of Health Informatics, Institute of Public Health, College of Medicine and Health Sciences, University of Gondar, Gondar P.O. Box 196, Ethiopia; 4Department of Epidemiology and Biostatistics, Institute of Public Health, College of Medicine and Health Sciences, University of Gondar, Gondar P.O. Box 196, Ethiopia; 5Centro de Pensamiento Medicamentos, Information y Poder, Universidad Nacional de Colombia, Bogotá 111321, Colombia

**Keywords:** COVID-19, vaccination, adverse events, LMICs

## Abstract

Background: Post-marketing vaccine safety surveillance aims to monitor and quantify adverse events following immunization in a population, but little is known about their implementation in low- and middle-income countries (LMICs). We aimed to synthesize methodological approaches used to assess adverse events following COVID-19 vaccination in LMICs. Methods: For this systematic review, we searched articles published from 1 December 2019 to 18 February 2022 in main databases, including MEDLINE and Embase. We included all peer-reviewed observational COVID-19 vaccine safety monitoring studies. We excluded randomized controlled trials and case reports. We extracted data using a standardized extraction form. Two authors assessed study quality using the modified Newcastle–Ottawa Quality Assessment Scale. All findings were summarized narratively using frequency tables and figures. Results: Our search found 4254 studies, of which 58 were eligible for analysis. Many of the studies included in this review were conducted in middle-income countries, with 26 studies (45%) in lower-middle-income and 28 (48%) in upper-middle-income countries. More specifically, 14 studies were conducted in the Middle East region, 16 in South Asia, 8 in Latin America, 8 in Europe and Central Asia, and 4 in Africa. Only 3% scored 7–8 points (good quality) on the Newcastle–Ottawa Scale methodological quality assessment, while 10% got 5–6 points (medium). About 15 studies (25.9%) used a cohort study design and the rest were cross-sectional. In half of them (50%), vaccination data were gathered from the participants’ self-reporting methods. Seventeen studies (29.3%) used multivariable binary logistic regression and three (5.2%) used survival analyses. Only 12 studies (20.7%) performed model diagnostics and validity checks (e.g., the goodness of fit, identification of outliers, and co-linearity). Conclusions: Published studies on COVID-19 vaccine safety surveillance in LMICs are limited in number and the methods used do not often address potential confounders. Active surveillance of vaccines in LMICs are needed to advocate vaccination programs. Implementing training programs in pharmacoepidemiology in LMICs is essential.

## 1. Key Points

Active surveillance studies have been used to monitor COVID-19 vaccine safety in low- and middle-income countries.Most studies were cross-sectional with limited outcome validation and no temporal assessment.Major vaccination data sources were medical charts or self-reported cases based on clinical signs or symptoms.Only one-third of the studies employed parametric models, such as logistic regression (*n* = 17, 29.3%) and Cox regression (*n* = 3, 5.2%).

## 2. Background

The rapid transmission of COVID-19 and its high death toll during the pandemic affected every aspect of daily life [1]. Lockdowns were mandatory worldwide to save lives, and the need for a COVID-19 vaccine was urgent. For vaccine development, extensive trials were performed and several vaccines received approval for emergency use in different countries and regions. The introduction of COVID-19 vaccines significantly altered the pandemic’s epidemiology by reducing morbidity [2,3,4,5]. However, like all other medicines, vaccines may cause adverse effects. This stimulates the interest of global scientific communities in studies on the safety monitoring of COVID-19 vaccines. Clinical trials, mostly phase I and II, showed only minor adverse effects [6,7]; however, their long-term effectiveness and safety are little known, particularly when used on a large scale. Post-licensure evaluation is needed to understand how the vaccines work in specific populations, scenarios, and real world settings [8]. Consequently, the World Health Organization (WHO), Global Alliance for Vaccines and Immunization (GAVI), and other international organizations provided technical, logistical, and financial support to procure COVID vaccines for low- and middle-income countries (LMICs) to control this pandemic [9,10].

Many regulatory agencies and public health organizations, such as WHO, European Medicines Agency (EMA), CDC, the U.S. Food and Drug Administration (FDA), and other national competent authorities (NCA), continue to monitor the safety of COVID-19 vaccines [11,12,13], but many NCAs have limited resources to support setting up infrastructures in low- and middle-income countries (LMICs) for safety monitoring. LMICs have restricted abilities for the active surveillance of vaccines and therefore, their data may not be adequate for regulatory decisions [14]. High-quality pharmacovigilance activities and pharmacoepidemiologic studies are required in specific geographies for the vaccines that were licensed there. Thus, there is a need to urgently understand how the safety evaluation of vaccines are done in LMICs [15,16,17].

Safety data are generated from several sources, such as spontaneous reports, clinical studies, and scientific literature. So far, a variety of study designs and analytical methods have been used to monitor the safety of COVID-19 vaccines in LMICs, yielding heterogeneous results and uncertainty about whether the observed differences are due to true variation between populations or caused by varying levels of confounding or bias related to the designs and methods used. However, these studies could contribute to vaccine safety in LMICs by detecting any potential adverse events associated with the use of the vaccines. Particularly, during the post-licensure phase and mass vaccination periods, simple and easily implementable study designs and methods may be preferred over more complex alternatives. Conversely, it is unclear which methods will achieve the best balance of speed and robustness [8,18,19,20,21,22,23,24,25,26]. Similar concerns were raised in the different safety data sources analyses, which employed statistical methods based on two-by-two contingency tables comparing a specific drug to other drugs and an adverse event (AE) to other AEs. To address these concerns, the safety and integrity of data must be prioritized to keep confidence in the vaccination process.

This study aimed to describe the methods and statistical approaches of COVID-19 vaccine safety studies that were performed in LMICs, including the types of data sources and methodological approaches used.

## 3. Methods

### 3.1. Search Strategy

We performed a systematic review of the literature to find peer-reviewed research studies on the safety of the COVID-19 vaccine to analyze the study design and methods for LMICs applicability. This systematic review was conducted according to the Preferred Reporting Items for Systematic Review and Meta-Analyses (PRISMA) statement [27]. This review focused on studies of adverse events following COVID-19 vaccination in LMICs published from 1 December 2019 up to 18 February 2022. For this review, the list of low- and middle-income countries was defined according to the World Bank classification, which is based on income [28]. Middle-income countries included both lower- and upper-middle-income countries. A systematic search was performed by Utrecht University’s librarian from the main medical databases, including Ovid MEDLINE and Embase.

The search terms were constructed using terms for the intervention of interest (COVID-19 vaccine) and the study design (observational studies: e.g., cohort, cross-sectional, and case-control studies). Medical subject heading (MeSH) terms, plain language, keywords, and synonyms were used, combined with “COVID-19” or “SARS-CoV-2” vaccines, observational study design, and low- and middle-income countries (Appendix A).

### 3.2. Eligibility Criteria

Primary, peer-reviewed original articles published in English were considered for inclusion. In addition, included studies were post-licensure studies and observational studies, such as cohorts (prospective and retrospective), case-control studies, test-negative designs, and screening studies; the study population had to be from LMICs and have received COVID-19 vaccination. Studies to be included had to report on vaccine safety monitoring, risk, safety assessment, and adverse events. Case reports, case series, randomized control trials (RCTs), and modeling studies were excluded. Reviews, editorials, letters, animal studies, commentaries, and conference abstracts were also excluded.

### 3.3. Data Extraction and Quality Assessment

Studies were independently screened on title and abstract, followed by full text by two independent reviewers (MMS and CMG), and disagreements were resolved by consensus or consultation with a third author (CED). Data extraction was performed by the lead author using a standardized data extraction template. The following data was collected for each study included: the date of publication, first author, country/ies, sample size, study design, adverse reactions, type of vaccines, the method for identification and validation of cases, and risk window of exposure. Moreover, we collected information on the analytic approaches used to estimate crude rate/odds ratios (data collection, characterizing methods of participants data, statistical model, subgroup/matching/sensitivity analysis, management of missing data and potential confounders), results (by the outcome of interest), and study limitations.

Safety outcomes were individual conditions/events following any COVID-19 vaccine, including local and systemic adverse reactions. Rare events were also considered.

### 3.4. Quality Assessment

Two reviewers (MMS and CMG) independently assessed the quality of each included study. We used the Newcastle–Ottawa Scale (NOS) to assess three broad perspectives: (1) the study groups’ selection, (2) comparability, and (3) the ascertainment of either the exposure for case-control studies or the outcome of interest for cohort studies [29,30]. All the included studies were assessed for clarity of aims, clear inclusion and exclusion criteria, reliability of outcomes, and an adequate description of data and adjustment for confounders, as well as proper use of a method and statistical analyses. The Newcastle–Ottawa Scale contains a maximum of 9 points which are based on: selection (up to 4 points), comparability (up to 2 points), and outcome (up to 3 points). Studies having NOS scores of 0–3, 4–6, and 7–9 were classified as low, moderate, and high quality, respectively.

### 3.5. Data Synthesis

Extracted data were qualitatively synthesized and we summarized the key findings in frequency tables and figures. The most reported safety parameters, statistical method suitability, and other findings were described in narrative form. Other data that were described in this literature review includes monitored vaccines, adverse event (AEs) studies, study design(s), data analysis approach, and the signal detection method that was employed.

## 4. Results

### 4.1. Study Selection 

About 4763 articles were found in the literature search. After removing duplicates, 4286 unique articles were considered for inclusion in the review (Appendix A). During the screening by title and abstract, 4040 articles were excluded. The remaining 246 full-text articles were assessed for inclusion and 58 articles met the inclusion criteria and were finally included in the analysis. Figure 1 shows details of this selection process.

### 4.2. Study Characteristics

The majority of the studies included in this review were conducted in middle-income countries, with 26 studies (45%) in lower-middle-income and 28 (48%) in upper-middle-income countries. More specifically, 14 studies were conducted in the Middle East region, 16 in South Asia, 8 in Latin America, 8 in Europe and Central Asia, and 4 in Africa. All studies were conducted between 2020 and 2022. The majority of studies [37 studies, (63.79%)] recruited high-risk group populations, such as healthcare workers, immunocompromised patients, or elderly persons. The median sample size of the studies was 442 subjects [interquartile range (IQR): 219, 830] (Table 1). Thirteen studies (22.41%) used cohort study and 41 were cross-sectional. Only two studies used comparator for safety assessment in which they compared Sinovac with Pfizer vaccines.

### 4.3. Methodological Quality

Overall, all cross-sectional studies had a score below 5 points, but only 8 of them scored 4 points. No disagreement between authors was seen.

Out of the fifteen cohort studies conducted in LMICs, two (3%) of the studies were rated as good (7–8 points) while six (10%) were rated as medium, according to the Newcastle–Ottawa Scale methodological quality (5–6 points) evaluation. Appendix A present the Newcastle–Ottawa Scale scores for each study and a summary of every item.

### 4.4. Vaccines Studied

Multiple platforms and brands of COVID-19 vaccines were assessed in the different studies, such as live, inactivated, and combined, for potential adverse events after receiving an emergency license. The AstraZeneca (AZ) adenovirus platform vaccine was checked in 29 studies, while the Sinovac inactivated virus vaccine was evaluated in 28 studies; another commonly studied vaccine was the Pfizer/BioNTech COVID-19 mRNA vaccine, which was evaluated in 18 studies (Table 2). In seven studies, any COVID-19 vaccine was studied without specifying platform or brand.

### 4.5. Characteristics of the Reported Safety Data

Most studies (51 articles, 87.9%) reported that they collected safety data using a combination of self-administered questionnaires during scheduled visits, as well as using online platforms (Appendix A), while a tenth of the studies (5 articles, 8.6%) used medical records only.

All the reviewed studies showed that they collected data on predefined local and systemic reactions (Table 3). About 5 studies reported positive tests for COVID-19 and other complications after vaccinations. Only a few studies reported adverse events with different severity levels, e.g., mild, moderate, and severe. Most of the reported risk windows ranged from 7 to 21 days (about 3 weeks) (Figure 2).

### 4.6. Study Designs Employed and Signal Detection Method

Most studies were cross-sectional studies (41 articles), collecting information on conditions and vaccination at a single point in time or retrospectively, while only a quarter of them were cohort studies (15 articles), including a retrospective cohort study without a control group. Over 50 studies (86.2%) did not perform outcome validation against diagnosis codes and did not use more clinical information such as medications, laboratory test results, and referrals to specialists to find cases. Only 8 (13.8%) articles confirmed outcomes against medical charts, such as the WHO causality assessment [31,32,33,34,35,36,37,38]. Nearly all 52 studies (94.8%) used medical charts or self-reported cases based on clinical signs or symptoms. Only three (5.2%) studies used spontaneous reporting, and two studies were based on surveillance registry.

### 4.7. Statistical Analysis for Safety Data Analysis

All the included articles used descriptive statistics as one of the methods to summarize AE (adverse events) frequencies. Mainly, proportions or counts were the descriptive statistics used in all studies to report safety data. Incidence rates were reported in 5 articles (8.6%). About 44 studies (75.9%) reported univariate inferential statistical methods, including Chi-square or Fisher’s exact test (40 studies; 70%), *t*-test (8 studies; 13.8%), ANOVA (5 studies; 8.6%), and nonparametric tests such as Mann–Whitney/Wilcoxon (12 studies; 20.7%). Around 20 studies reported multivariate statistical methods, including binary regression (*n* = 17, 29.3%), survival analysis like Cox regression, and Kaplan–Meier curves (*n* = 3; 5.2%). In terms of results presentation, 58 studies (100%) narratively presented their safety results. Most of them (57 studies; 98.3%) presented results in tabular format, from which 55 studies reported p-values to test for treatment effects and point estimates with their respective confidence intervals. A total of 34.5% used more than one inferential method and no study reported methods for handling missing data (Table 4 and Appendix A).

## 5. Discussion

This review summarizes the study designs and methods used for adverse events assessment following COVID-19 vaccination in low- and middle-income countries (LMICs). Our review found that studies in LMICs relied on active surveillance and were limited in terms of causal inference. Most studies were conducted among healthcare workers and people with underlying conditions. This may be due to the accessibility of participants and data collection. Moreover, healthcare workers were the first vaccinated group as they were at a higher exposure risk of COVID-19 because they participated in routine screening activities undertaken within the health system [39]. Additionally, people that were immunocompromised were prioritized for vaccination. This gives the opportunity to collect safety data for these populations easily and early in the pandemic for research purposes. For this reason, there is a high interest to consider these populations when designing vaccine safety research.

In our review, common adverse events were reported with mild or moderate severity and serious adverse reactions, which is consistent with other medical literature [40,41,42,43]. This finding is also in line with clinical trials and global cohort event monitoring studies, which showed that injection site or local reactions are more often reported than systemic reactions and serious adverse events [44,45,46,47]. Moreover, according to literatures [6,7,40,48,49], the most known side effects of COVID-19 vaccines are mild and temporary, but they should be continuously monitored and evaluated. On the other hand, another review’s data indicated that information in children’s safety is generally scarce due to an ethical dilemma for COVID-19 immunization [50,51]. Additionally, vaccine injury compensation programs are limited among countries, with most of them concentrated in Europe and the United States [52,53,54].

Our study review has found methodological issues for improvement when designing a study of vaccine safety monitoring in LMICs. These include the use of proper study design and statistical methods that enable causal inference, inclusion of a generalizable study population and a comparator group, outcome validation, exposure assessment, and ensuring sufficiently long follow-up time. The most common study design found in our review was a cross-sectional design, which may have been driven by the absence of large, reliable, and interlinked databases in low- and middle-income countries. Cohort or retrospective studies were the second most used study design, but there were challenges in prospectively enrolling a suitable comparator group [15,55,56]. However, more data are needed from observational studies to inform decision makers and to obtain a more comprehensive view of vaccine safety in the perspective of diverse population and settings. This is supported by vaccine safety method guidelines [8,20]. In line with this, many studies that have synthesized COVID-19 vaccine safety estimates from observational studies have been used to support countries in the aftermath of COVID-19 vaccination, in addition to clinical trials. In particular, cohort, case-control, and self-controlled studies have been employed often in the evaluation of the COVID-19 vaccine [57,58] but they may pose methodological challenges. To mention examples, there are challenges to obtain information on the size of the population at risk and to identify an appropriate comparison group. As a result, there is a call to revolutionize the research landscape of COVID-19 vaccine safety monitoring in LMICs that have the potential to significantly improve scientific discoveries [59].

Regarding the statistical analysis methods used, standard statistical tests were used to examine safety data. The most reported methods that allowed adjustment for confounders were logistic regression and Cox regression analysis [32,49]. The choice of specific statistical tests should be guided by the aims, study designs, and outcomes. Some of the studies included showed methodological flaws, thereby introducing a substantial risk of biases. For instance, they only included the subjects at high risk for serious COVID-19 infections or outcomes collected retrospectively, which could introduce high misclassification and recall biases. Furthermore, only few studies performed statistical analysis with adjustment for potential confounders [32,49,60,61,62,63,64,65]. Many studies suggested new advanced methods and study designs to minimize the impact of confounders, such as self-controlled designs [18,58] which automatically address time-invariant confounders. We argue combined data sources using innovative data modeling approaches would be globally imperative [66,67,68]. We strongly advise researchers to adhere to the reporting standards, such as REporting of studies Conducted using Observational Routinely collected health Data (RECORD) [21], so that readers can easily grasp the methodology and research findings.

This review emphasizes the importance and need of well-performed observational studies to support vaccine safety surveillance in LMICs. Although COVID-19 vaccination is considered one of the most important strategies to curb a pandemic [69], adverse events could be a barrier to people’s willingness to be vaccinated. Some LMICs have already been affected by low vaccine supply, lower vaccination rates, and limited resources. This review also illustrates a lack of large data sets from electronic sources in LMICs, such as electronic health records, claims data, or patient registries, as well as observational data used to investigate new vaccines in routine practice.

Based on this review, the studies conducted in LMICs to investigate the safety of vaccines have some limitations. Firstly, many studies were cross-sectional, which means they were conducted at a single point in time and did not follow up with participants over an extended period. We argue that longitudinal studies that prospectively follow participants over time can supply more robust evidence for vaccine effectiveness and safety. Secondly, the studies relied on surveys, which can be subject to biases and inaccuracies. Use of more objective measures, such as laboratory tests or medical records, would generate stronger evidence. Thirdly, while a wide variety of vaccines were investigated, most studies did not have a non-vaccinated or other active comparator group. This makes it challenging to figure out whether the vaccine was genuinely effective or if other factors could explain any observed effects and whether adverse events reported were indeed more common after vaccination. The findings suggest that active surveillance studies are possible in LMICs but that there is room for improvement in their quality. VAC4EU in Europe and SPEAC in the United States are good examples of robust surveillance systems for vaccine monitoring in high-income countries (HICs). Lastly, findings from this study should be interpreted with caution since some studies might have been missed since we only included articles published in English.

Strong collaborations between health regulators and the scientific community could ensure vaccine availability and use in LMICs. Designing and conducting safety monitoring studies in specific settings and populations can provide benefits. This report does not address COVID-19 prevention in specific populations, such as younger adolescents, children, and pregnant women. We noted that observational studies are needed to generate the necessary information to initiate development and combat pandemics in the current expansion of other infectious disease outbreaks. Given the expected scope of the COVID-19 vaccine rollout and the number of vaccine products, developing new data systems or improving and extending existing ones will be critical. Furthermore, existing population-based cohorts may be used for subgroup analyses and confounder adjustment through data linkage and leverage to reduce biases associated with observational study designs. To complement, data mining approaches can be employed to look for the possible associations between receiving COVID-19 vaccines and any of a wide range of medically attendant adverse events using spontaneous reporting systems.

Our findings highlight the importance of strong pharmacovigilance systems, particularly in low-and middle-income countries. Improving the quality of data and research in these settings is critical for understanding the impact of vaccines and other interventions on public health outcomes. Efforts to support local researchers in building their capacity are essential to this goal. International and south-south collaborations should be fostered to ensure that the best practices and lessons learned from one setting can be applied to others. In addition, ongoing monitoring and evaluation of vaccine effectiveness and safety, particularly in the context of emerging variants, will be critical in the coming years. Overall, the COVID-19 pandemic has underlined a wakeup call for the need to invest in surveillance infrastructure and research systems. We should seize this opportunity to build a more equitable and resilient future for all.

## 6. Conclusions

In conclusion, observational studies can generate valuable information on vaccine safety and effectiveness in LMICs. However, efforts must be made to improve the quality and use of rigorous methods. Supporting and training local epidemiologists is critical to achieving this goal and developing long-term research development in LMICs. It is also important to note that vaccine safety and effectiveness monitoring should be an ongoing process, and LMICs should have the resources and infrastructure in place to conduct continuous surveillance of vaccines and other interventions. This will be critical in the context of emerging infectious diseases and potential future pandemics. Overall, investing in surveillance infrastructures, global health and pharmacoepidemiologic research in LMICs is critical to improve health outcomes and achieving greater equity.

## Figures and Tables

**Figure 1 vaccines-11-01035-f001:**
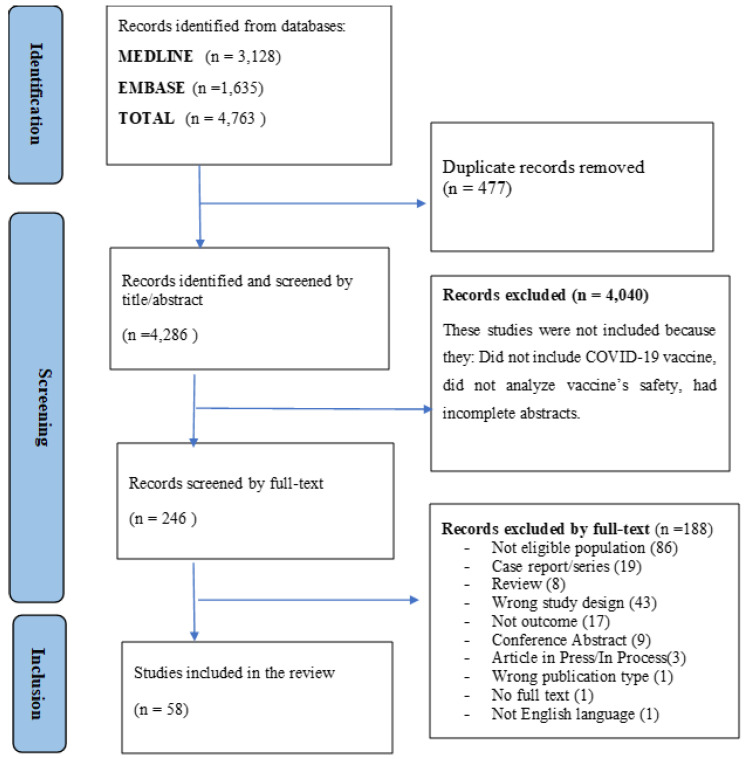
PRISMA flow diagram of the study selection process.

**Figure 2 vaccines-11-01035-f002:**
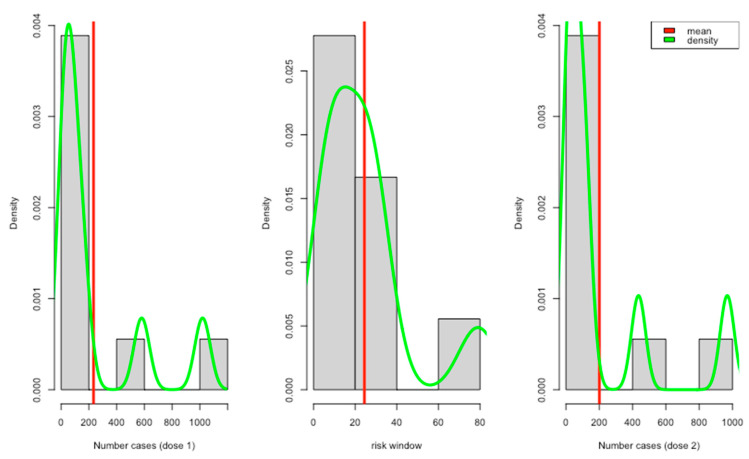
Report case distribution per dose with risk windows.

**Table 1 vaccines-11-01035-t001:** Summary characteristics of included studies.

Study Characteristics	Classification	Number (%)
Study Designs	Cross-Sectional Studies/Descriptive studies	41 (70.69)
	Cohort Studies	13 (22.41)
	Retrospective	2 (3.45)
	Both Cross-sectional and Cohort	1 (1.72)
	Cross-sectional—Sequential mixed-method	1 (1.72)
Country world bank classification	Low-income economies	4 (7.00)
	Lower-middle-income economies	26 (45.00)
	Upper-middle-income economies	28(48.00)
Data sources	Primary data	51 (87.93)
	Secondary data	5 (8.62)
	Mixed	2 (3.45)
Source of vaccination data	Spontaneous reporting	3 (5.17)
	Registry in Epidemiological Surveillance System	2 (3.45)
	Self-reported (Primary data collection)	52 (89.66)
	Active surveillance	1 (1.72)
Populations of interest	High-risk population (e.g., healthcare workers, immunocompromised hosts)	37 (63.79)
	Children	1 (1.72)
	Adults	15 (25.86)
	All group	5 (8.62)
Analysis method	Statistical tests (association)—No adjustment for confounder	47 (82.46)
	Advanced modeling (e.g., regression analysis)—Adjustment for confounders	10 (17.54)
Study type	Near real-time surveillance	57 (98.28)
	Phase IV observation study	1 (1.72)
Comparator for safety assessment (e.g., non-exposed, active comparator/vaccine)	Yes	2 (3.45)
	No	56 (96.55)

**Table 2 vaccines-11-01035-t002:** Type of vaccines studied by the selected studies.

Manufacturer	Name of Vaccine	Platform	Frequency
AstraZeneca, AB or Serum Institute of India Pvt. Ltd., Maharashtra, India	AZD1222 Vaxzevria or Covishield (ChAdOx1_nCoV-19)	Recombinant ChAdOx1 chimpansee adenoviral vector	29
Sinovac Life Sciences Co., Ltd., Hong Kong, China	COVID-19 Vaccine (Vero Cell), Inactivated/ CoronaVac (Sinopharm or Sinovac or CoronaVac)	Inactivated virus	28
BioNTech Manufacturing GmbH, Mainz, Germany	BNT162b2/COMIRNATY Tozinameran (INN)	Nucleoside modified mRNA	18
Russian Direct Investment Fund, Moscow, Russia	Sputnik V	Human Adenovirus Vector	10
Any type of Vaccine	Any (Not specified)	N/A (Not mentioned)	7
Bharat Biotech, Telangana, India	SARS-CoV-2 Vaccine, Inactivated (Vero Cell)/ COVAXIN	Whole-Virion Inactivated	4
CasinoBio Cansino Biologics, Tianjin, China	Ad5-nCoV, Convidecia	Recombinant Novel Coronavirus Vaccine (Adenovirus Type 5 Vector)	4
Moderna Biotech, Cambridge, MA, USA	mRNA-1273, Spikevax	Nucleoside modified mRNA	3
Janssen–Cilag International NV, Beerse, Belgium	Ad26.COV2.S, JCOVDEN	Recombinant, replication incompetent adenovirus type 26 (Ad26) vectored vaccine encoding the (SARS-CoV-2) Spike (S) protein	1

Note: Entries in this table are not mutually exclusive and do not add up to the total 58 included articles since some articles used more than one approach.

**Table 3 vaccines-11-01035-t003:** Adverse events reported by the studies.

Adverse Events Category	Number of Studies	Adverse Events
Systemic event reactions	53	Fever or hyperthermia or feverish, headaches, fatigue, vomiting, diarrhea, muscle pain, joint pain, cough, nausea, dyspnoea, appetite impaired, dizziness, mucosal abnormality, pruritus, hypersensitivity, syncope, asthenia, rhinorrhoea, malaise, sore throat (throat irritation), pain in the oropharynx (pharyngalgia), hives, nasal congestion.
Injection site adverse reactions	53	Pain, induration, redness, or erythema, swelling, itch, muscular weakness.
Serious vaccine-related adverse event	3	Deaths, hospitalization, thrombotic complications.
Others	5	Reported positive test for COVID-19 and other complications
Doses	58	Investigated Dose 1 effects
	12	Both dose (1 & 2)
	1	Booster analysis
Outcome identification methods and validation (by diagnostic codes, …)	8	Common Terminology Criteria for Adverse Events (CTCAE) version V5 World Health Organization-Uppsala Monitoring Centre (WHO-UMC) causality assessment scale IgG anti-spike-protein antibodies test and laboratory tests Medically reviewed at in-person visits

**Table 4 vaccines-11-01035-t004:** Statistical approaches used for safety data analysis.

Statistical Method/Approach		Number of Articles (%)
Descriptive statistics	Proportion/count	57 (98.3)
	Mean/median	27 (46.6)
	Incidence rate	5 (8.6)
Inferential methods	Univariate methods	44 (75.9)
	Fisher’s exact test/Chi-square	40 (70)
	Mann–Whitney/Wilcoxon	12 (20.7)
	*t*-test	8 (13.8)
	ANOVA	5 (8.6)
Multivariable modeling	Binary regression	17 (29.3)
	Survival Analysis	3 (5.2)
Number of inferential methods	One method	37 (63.8)
	More than one method	20 (34.5)
	None	1 (1.7)
Analysis approach	Handling missing data	0 (0)
	Imputation	0 (0)
	Model diagnostics and validity checks (e.g., goodness of fit, identification of outliers, and co-linearity)	12 (20.7)
Results presentation	Tables	57 (98.3)
	Point estimate and confidence interval	55 (94.8)
	p value	55 (94.8)
	Graphs	7 (10.3)

Note: Entries in this table are not mutually exclusive and may not add up to the total of 58 included articles since some articles used more than one approach.

## Data Availability

All data are provided in the article or Appendix A.

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
