# Peer review of "COVID-19 Vaccine Safety Monitoring Studies in Low- and Middle-Income Countries (LMICs)—A Systematic Review of Study Designs and Methods"

_vaccines, 2023, doi:10.3390/vaccines11061035_

Round 1

Reviewer 1 Report

The manuscript methodically addresses the issue of post-marketing vaccine safety surveillance. The methods are comprehensively described, as are the results. In view of the relevance of the topic, a brief discussion on compensation for vaccine damage could be useful; I suggest consulting and citing some references:

Assadi M, Kiani M, Shamsi Gooshki E, Aryanian Z, Afshar ZM, Hatami P. COVID-19 vaccination in children as a global dilemma through an ethical lens: A retrospective review. Health Sci Rep. 2022 Dec 3;6(1):e976. doi: 10.1002/hsr2.976. PMID: 36479386; PMCID: PMC9719287.

Beccia F, Rossi MF, Amantea C, Villani L, Daniele A, Tumminello A, Aristei L, Santoro PE, Borrelli I, Ricciardi W, Gualano MR, Moscato U. COVID-19 Vaccination and Medical Liability: An International Perspective in 18 Countries. Vaccines (Basel). 2022 Aug 7;10(8):1275. doi: 10.3390/vaccines10081275. PMID: 36016163; PMCID: PMC9415029.

Crum T, Mooney K, Tiwari BR. Current situation of vaccine injury compensation program and a future perspective in light of COVID-19 and emerging viral diseases. F1000Res. 2021 Jul 26;10:652. doi: 10.12688/f1000research.51160.2. PMID: 35035888; PMCID: PMC8733825.

D'Errico S, Zanon M, Concato M, Peruch M, Scopetti M, Frati P, Fineschi V. "First Do No Harm". No-Fault Compensation Program for COVID-19 Vaccines as Feasibility and Wisdom of a Policy Instrument to Mitigate Vaccine Hesitancy. Vaccines (Basel). 2021 Sep 30;9(10):1116. doi: 10.3390/vaccines9101116. PMID: 34696224; PMCID: PMC8540114.

Minor editing of English language required

Author Response

Dear Reviewer,

We appreciate for taking time to carefully review the manuscript and give detailed constructive comments, which have greatly helped to improve this paper. Please find our point by point response to each comment.

With regards

Reviewer 2 Report

 The article appears to provide a background on the urgent need for COVID-19 vaccines and the importance of monitoring their safety in low- and middle-income countries (LMICs). The article highlights the challenges faced by LMICs in setting up infrastructures for active surveillance of vaccines, and the potential limitations of their data for regulatory decisions.

The authors also mentions the various study designs and analytical methods used to monitor the safety of COVID-19 vaccines in LMICs, and the need for simple and easily implementable study designs and methods during the post-licensure phase and mass vaccination periods. The article aims to describe the methods and statistical approaches used in COVID-19 vaccine safety studies performed in LMICs, including the types of data sources and methodological approaches employed.

Overall, this article provides a clear overview of the main themes and objectives of the article, which appears to be focused on the methodological approaches used to assess adverse events following COVID-19 vaccination in LMICs.

The following reference which should be included in the review

-COVID-19 vaccine safety monitoring in low and middle income countries
Walter Straus, Harvey Rubin

Safety and Efficacy of the BNT162b2 mRNA Covid-19 Vaccine

Fernando P. et al., 2020

http://refhub.elsevier.com/S0264-410X(22)00822-2/h0010

http://refhub.elsevier.com/S0264-410X(22)00822-2/h0015

Author Response

Dear Reviewer,

We appreciate for taking time to carefully review the manuscript and give detailed and constructive comments, which have greatly helped to improve this paper. Please find attached below is our point-by-point response to each comment.

With regards,

Reviewer 3 Report

Please see review letter.  Suggestions are shown in italics.

Author Response

Dear Reviewer,

We appreciate for taking time to carefully review the manuscript and give detailed and constructive comments, which have greatly helped to improve this paper.

Please find attached below our point-by-point response to each comment.

With regards,
